# Eliciting preferences for outpatient care experiences in Hungary: A discrete choice experiment with a national representative sample

Óscar Brito Fernandes[1,2]*, Márta Péntek[1], Dionne Kringos[2], Niek Klazinga[2], László Gulácsi[1], Petra Baji[1]

1 Department of Health Economics, Corvinus University of Budapest, Budapest, Hungary, 2 Department of Public and Occupational Health, Amsterdam UMC, University of Amsterdam, Amsterdam Public Health Research Institute, Amsterdam, The Netherlands

* obritofernandes@uni-corvinus.hu

**Data Availability Statement:** All relevant data are within the paper and its Supporting Information files.

## Abstract

### Introduction

Patient-reported experience measures (PREMs) are central to inform on the responsiveness of health systems to citizens' health care needs and expectations. At their current form, PREMs do not reflect the weights that patients assign to varying aspects of the care experience. We aimed to investigate patients' preferences and willingness to pay (WTP) for attributes of the care experience in outpatient settings.

### Methods

A discrete choice experiment was conducted among a representative sample of the general adult population of Hungary (n = 1000). Choice set attributes and levels were defined based on OECD's standardized PREMs (e.g. a doctor spending enough time in consultation, providing easy to understand explanations, giving opportunity to ask questions, and involving in decision making) and a price attribute. Conditional and mixed logit analyses were conducted. WTP estimates were computed in preference and WTP space.

### Results

The respondents most preferred attribute was that of a doctor spending enough time in consultation, followed by involvement in decision making. Moreover, waiting times had a less important effect on respondents' choice preference compared with aspects of the doctor-patient relationship. Estimates in the WTP space varied from €4.38 (2.85–5.90) for waiting an hour less at a doctor's office to €36.13 (32.07–40.18) for a consultation where a doctor spends enough time with a patient relative to a consultation where a doctor does not.

### Conclusions

A preference-based PREMs approach provide insight on the value patients assign to different aspects of their care experience. This can inform the decisions of policy-makers and

**Funding:** This research was funded by the Higher Education Institutional Excellence Program of the Ministry of Human Capacities in the framework of the 'Financial and Public Services' research project (20764-3/2018/FEKUTSRTAT) at Corvinus University of Budapest. The research was developed within a Marie Skłodowska-Curie Innovative Training Network (HealthPros — Healthcare Performance Intelligence Professionals) that has received funding from the European Union's Horizon 2020 research and innovation programme under grant agreement Nr. 765141 (https://healthpros-h2020.eu). The funders had no role in study design, data collection and analysis, decision to publish, or preparation of the manuscript.

**Competing interests:** In connection with writing this article, OBF, MP, LG and PB received grant support from the Higher Education Institutional Excellence Program of the Ministry for Innovation and Technology in the framework of the Financial and Public Services research project (NKFIH-1163-10/2019) at Corvinus University of Budapest.

other stakeholders to coordinate efforts and resource allocation in a more targeted manner, by acting on attributes of the care experience that have a greater impact on the implementation of patient-centered care.

## Introduction

Many health care systems across Europe are committed to further improve responsiveness to citizens' health care needs and expectations [1]. A cornerstone of this growing people-centered culture is that of empowering and engaging citizens to undertake an active role on their health care management. Within a value-based health care framework, a rising approach to activate citizens' voice into health systems performance assessment is that of considering the experiences of care of patients [2, 3].

In the last decade, we have observed a growing interest on patient-reported experience measures (PREMs). These measures became a widely used quality indicator to inform the general public, policy-makers, but also health care professionals and organizations about patient-centered health care service delivery, wherein aspects of the care experience are measured [4, 5]. However, a shortcoming to most instruments on PREMs should be acknowledged: many lack standardization or proper reporting about its validity/reliability [4].

The Organisation for Economic Co-operation and Development (OECD) has been advocating for data collection on patient experiences across its member-states [1, 6]. To gauge aspects of the care experience in outpatient settings, the OECD endorses a standardized set of questions, following earlier efforts of the Commonwealth Fund (e.g. International Health Policy Survey) and the Agency for Healthcare Research and Quality (e.g. the program Consumer Assessment of Healthcare Providers and Systems). These PREMs focus on patient-centeredness features, such as those of the patient-doctor communication and patient involvement in decision making.

The Hungarian health system is organized around a single health insurance fund, which provides health coverage for nearly all residents. However, the benefit package is less comprehensive than in most European Union (EU) countries, and thus, a large number of people rely on out-of-pocket payments to access care [7]. Public health expenditure accounts for two-thirds of the total health expenditure, which sets the levels of out-of-pocket payments to almost double of the EU average (27% vs 16%) [7]. Out-of-pocket payments have been increasing partly because of the rising co-payments with pharmaceuticals and outpatient care, growing utilization of care providers in the private sector and the prevalence of informal payments [8]. Given this context, citizens' experiences of care may be undermined up to some extent.

In Hungary, recent applications of the OECD's set of recommended PREMs are detailed in two articles: one sought to measure experiences of care in outpatient settings [9]; and the other focused on unmet health care needs due to cost and difficulties in travelling [8]. However, to measure those aspects of the care experience is only a first key layer in developing more targeted patient-centered policies. Given that patient experiences are much influenced by one's perceptions and social representations of what quality care is, different patients may value certain aspects of their experiences more than others [10]. And at this moment, PREMs fail to reflect the preference weights that patients assign to varying aspects of the care experience.

Thus, the aim of this study is to examine the weights that patients assign to attributes of the care experience included in the OECD's set of recommended PREMs. Moreover, we compute the willingness to pay for improvement on attributes of the care experience (which also reflect the preference weight attached to those attributes).

To achieve our aim, we use a discrete choice experiment (DCE) technique. The DCE is a stated preference method very popular in the field of health economics [11], whereby respondents are confronted with at least 2 hypothetical scenarios (e.g. medical consultations with different characteristics) of which they have to choose one. Each scenario is composed with varying levels of different attributes (e.g. aspects of the care experience). The DCE results can inform on the preference weights that respondents assign to attributes. By combining information on both experiences and preferences of patients, further intelligence can be synthesized to assist policy-makers and other key-stakeholders on the development of tailored patient-centered policies.

## Methods

### Attribute selection

The attribute selection for aspects of the care experience that add value to patients was based on the OECD's proposed set of questions to gauge patient-reported experience measures (PREMs) in outpatient settings [6]. Following best practices of attribute identification and selection [12], we chose those PREMs because of several reasons: 1) a recent systematic review identified those measures as common in DCE studies to elicit patients' preferences for primary health care [13]; 2) previous research has identified strong linkages between those attributes and quality of care, clinical safety and effectiveness [14, 15]; 3) those attributes are strong predictors of one's perception of quality of an outpatient consultation [16], which may be an important consideration when choosing a consultation and; 4) those attributes represent a balance between what is relevant for patients and the health policy context [1, 6]. Attributes covered aspects such as those of people's access to care (e.g. waiting time for an appointment and waiting time at a doctor's office) and experiences with outpatient care. About the latter, the attributes focused on aspects of care such as those of a doctor spending enough time with a patient, providing easy to understand information, giving a patient opportunity to ask questions or raise concerns about recommended treatments, and involving a patient in decision making about care and treatment. Additionally, we used a price attribute (out-of-pocket payments) to compose each outpatient consultation scenario (Table 1).

### Attribute levels selection

The attribute levels were selected based on the original scale of OECD's instrument as follows: for waiting times, we considered a restrained number of options that covered the full range of answer options of the original question; for attributes of the care experience we grouped the original 4-point Likert scale response options to a binary answer option (negative or positive experience on that attribute). These adaptations were needed to keep the cognitive complexity of the choice tasks at a reasonable level for respondents. Regarding the price attribute, we adopted price levels that cover well those of real-life settings in the Hungarian context, both in public and private practices. Overall, we used 7 attributes (3 with 4 levels and 4 with 2 levels) which intended to be realistic, relatable, and understandable for respondents, but also to policy-makers.

### DCE tasks and experimental design

The DCE module started with a brief explanation about what was expected from the respondents regarding the choice tasks (S1 File). Afterward, respondents were instructed the following: "Imagine that you have a health problem that concern you but does not require immediate care and to receive health care you will be visiting a specialist for a consultation or

**Table 1. Attributes and levels used in the discrete choice experiment.**

| Attributes | Levels |
|---|---|
| $A_1$: Waiting time for an appointment | You have a medical appointment the **next day.** |
| | You have a medical appointment in **2 weeks.** |
| | You have a medical appointment in **6 weeks (1.5 months).** |
| | You have a medical appointment in **12 weeks (3 months)**. |
| $A_2$: Waiting time at the doctor's office | On the actual day of the consultation, you **do not have to wait** before you are actually seen. |
| | On the actual day of the consultation, you have to **wait 1 hour** before you are actually seen. |
| | On the actual day of the consultation, you have to **wait 2 hours** before you are actually seen. |
| | On the actual day of the consultation, you have to **wait 4 hours** before you are actually seen. |
| $A_3$: Doctor spending enough time in consultation | The doctor **does not spend enough time** with you during the consultation. |
| | The doctor **spends enough time** with you during the consultation. |
| $A_4$: Doctor providing easy to understand explanations | The doctor explains things in a way that **is not easy to understand**. |
| | The doctor explains things in a way that is **easy for you to understand.** |
| $A_5$: Doctor giving opportunity to ask questions/raise concerns | The doctor **does not give you an opportunity** to ask questions or raise concerns about recommended treatment. |
| | The doctor **gives you an opportunity** to ask questions or raise concerns about recommended treatment. |
| $A_6$: Doctor involving the patient in decision making about care/treatment | The doctor **does not involve you as much as you wanted** to be in decisions about your care and treatment. |
| | The doctor **involves you as much as you wanted** to be in decisions about your care and treatment. |
| $A_7$: Out-of-pocket payment | The consultation costs you **HUF 0 (0 Eur).** |
| | The consultation costs you **HUF 5 000 (15.73 Eur).** |
| | The consultation costs you **HUF 15 000 (47.18 Eur)**. |
| | The consultation costs you **HUF 30 000 (94.37 Eur).** |

Average currency conversion for February 2019: € 1 = HUF 317.91.

an examination." Next, respondents were asked to choose between two different outpatient consultation scenarios (*A* or *B*). All the tasks that were presented to the respondents for preference elicitation included all attributes, i.e. each consultation scenario was presented as full profile. We did not incorporate an opt-out or a status quo option, given that in the task instructions provided to respondents we assumed that they would have to seek care because of a concerning health problem at some point in time. Although an opt-out option could have reduced bias in parameter estimates, given that in real market scenario patients can opt-out of care or delay care, a forced choice method may lead to more thoughtful responses [17]. In addition, a *status quo* option was not included because this study aimed to estimate trade-offs between characteristics of a medical consultation (e.g. a doctor spending enough time in consultation with a patient or providing easy to understand explanations) rather than the expected uptake of certain consultations.

Considering the number of attributes and their levels, hypothetically respondents had to choose from 1024 different combinations. For the study to become feasible, we defined a D-efficient fractional design with priors set to zero, for main effects only, with adequate level balance and minimum overlap of attribute levels. We used Stata's *dcreate* command to maximize the D-efficiency of the design based on the covariance matrix of conditional logit model. The

design consisted of 20 choice sets sorted into 4 blocks, each with 5 tasks (S2 File). Blocks were randomly allocated to respondents (i.e. each respondent was faced with 5 choice tasks). By doing so, we expected to decrease respondents' fatigue while answering the choice tasks of the DCE and preserve the precision and reliability of the estimations.

## Preference elicitation

Respondents were asked to choose between two different outpatient consultation scenarios (*A* or *B*). The DCE was built in an unlabeled and forced choice format, i.e. the choice alternatives were not specified with any label and were only characterized by its attributes. The DCE module ended with two closed questions. First, a 7-point Likert scale on the degree of difficulty in answering the choice tasks (1: It was not difficult to 7: It was very difficult). Second, a question with multiple answer options on aspects that may have contributed to the revealed degree of difficulty (difficult to understand the different medical scenarios; difficult to imagine the need for medical care; difficult to choose between the two scenarios; difficult to interpret the description of medical treatment in the two scenarios; other reasons).

## Instrument design

The survey *Patient experiences in health care* consisted of three main modules: 'eHealth literacy', 'Shared decision-making' and 'Patient-reported experience measures', which are detailed in full elsewhere [8, 9, 18, 19]. The latter included a discrete choice experiment (DCE) to elicit respondents' preferences for aspects of the care experience in outpatient settings. All the choice tasks featured in the survey were mandatory, hence the full sample of 1000 respondents completed the DCE module (S1 Dataset). The attributes of the care experience were designed based on a set of standardized PREMs recommended by the OECD [6]. The survey was conducted in the Hungarian language; thus, a translation process of the PREMs questions was conducted, as described elsewhere [9].

We included 17 respondents for the pilot testing of the DCE. Most respondents were university students aged 18 and over that had an outpatient consultation in the previous 12 months (except dental care). The objective of this pilot testing was to detect for possible errors in the DCE module and to assess respondents' understanding on the choice tasks (including attributes and attribute levels). Feedback from the pilot testing suggested that the choice tasks were understandable and feasible for respondents. Only the following revision was made: to include an "other reasons" answer option to the question "Why was it difficult to answer to the questions?" This question was asked after the choice tasks to account for reasons that may have contributed to the difficulty in answering the choice tasks, besides those previously listed (*difficult to understand the different medical scenarios*; *difficult to imagine the need for medical care*; *difficult to choose between the two scenarios*; *difficult to interpret the description of medical treatment in the two scenarios*). No other revision was made to the DCE section of the survey.

## Data collection

Data were collected in early 2019 via an online self-administered survey from a panel of an internet survey company (Big Data Scientist Kft.). The recruitment process aimed at a target sample size of 1000 respondents based on rule of thumb [20]. A disproportionate stratified random sampling was employed to reflect the characteristics of the general adult population of Hungary in terms of sex, age (by age group: 18–24, 25–34, 35–44, 45–54, 55–64 or 65 and over years), highest education level attained (primary, secondary or tertiary), type of settlement (Budapest, other cities or village) and region of residence (Central, Eastern or Western Hungary). Given that this was an online survey and that the use of the internet is lower among

people aged 65 and older, the sampling aimed to reflect a fair representativeness of older age groups, in comparison with the distribution of older age strata in the general adult Hungarian population. We used publicly available information of the Hungarian Central Statistical Office to characterize the distribution of the general adult population [21].

Ethical approval to conduct this study was granted by the Medical Research Council of Hungary (Nr. 47654-2/2018/EKU). Respondents provided their informed consent at two moments: first, prior answering the questionnaire; second, at the time of submission. No personal identifying information was collected. The answers of respondents were anonymized prior to analysis.

## Statistical analysis

We used absolute and relative frequency to summarize the socio-demographic characteristics of the respondents.

DCE data were analyzed to predict choice and estimate preference weights via an indirect utility function. Following the random utility framework [22], the underlying utility that a respondent $n$ assigns to alternative $j$ can be written as $U_{jn} = V_{jn} + \varepsilon_{jn}$, where $V_{jn}$ is the deterministic component of utility. This component was defined by a vector of alternative-specific constant and a vector of attributes of the choice alternative $j$. We assumed that given two scenarios ($A$ and $B$), a respondent will have chosen alternative $A$ if $U_{An} > U_{Bn}$. Thus, we assumed that respondents were able to make trade-offs between attribute levels to maximize utility. We estimated the utility function as follows:

$$U_{jn} = \beta_0 + \sum_{k=1}^{7} \beta_k A_k + \varepsilon_{jn}$$

where $\beta_0$ is an alternative-specific constant that indicates respondents' preference weight for consultation $B$, and $\beta_1$ to $\beta_7$ represent the preference weight of each attribute level (compared to its reference level). Attributes on waiting times ($A_1$ and $A_2$) and out-of-pocket payment ($A_7$) were modelled as continuous; remaining attributes were dummy-coded (1: positive experience of care). We assumed errors to be independent and identically distributed following a type-one extreme value distribution. This specification resulted in a conditional logit (model 1) to estimate respondents' preferences. This parsimonious model assumed that all respondents have the same preferences, i.e. shorter waiting times, positive experiences of care and lower out-of-pocket payments. Given this unrealistic assumption, to account for preference heterogeneity, we also analyzed the data with mixed logit models considering 1000 Halton draws out of the sample per respondent (model 2 and 3).

In model 2, the alternative-specific coefficient and the out-of-pocket payment ($A_7$) coefficients were specified to be fixed; waiting time coefficients ($A_1$ and $A_2$) were specified to be lognormally distributed (assuming every respondent is likely to prefer shorter waiting times) whilst other attributes were specified as having a random component normally distributed. Both model 1 and 2 were included as benchmark model specifications, where the latter is still quite common in the DCE literature because it allows the computation of willingness to pay estimates in the preference space in a straightforward manner [23]. To improve the realism of model assumptions, in model 3 we have also specified the out-of-pocket coefficient to be lognormally distributed allowing the preference for this attribute to vary across respondents.

## Willingness to pay

Willingness to pay (WTP) assigns monetary value to attributes (i.e. how much money were respondents willing to pay for a one-unit improvement in one of the attribute levels).

Estimates for WTP in preference space may be computed as the ratio of the coefficient for an attribute and the out-of-pocket payment coefficient. However, this approach may result in highly skewed WTP distribution. Hence, we computed model 4 in WTP space [23], following the specification of model 3 in preference space. Waiting time and out-of-pocket payment were entered as negative because of the log-normal distribution assumption. Confidence intervals were estimated with the delta method.

For presentation purposes, we converted the out-of-pocket payment currency from Hungarian Forint (HUF) to Euro (€). We considered the average currency exchange rate of the European Central Bank by the time of data collection (February 2019): € 1 = HUF 317.91.

All statistical analyses were performed in Stata (version 16) with the user-written *mixlogit* [24] and *mixlogitwtp* [25] modules to compute mixed logit model estimates in preference and WTP space.

# Results

## Respondents' characteristics

A total of 1000 questionnaires were completed (Table 2). Women represented 55% of the respondents. Average age was 46 years old (standard-deviation: 18) and most respondents' age was between 35–64 years old (46.3%). More than 70% of the respondents completed secondary

**Table 2. Socio-demographic characteristics of the respondents.**

| | Sample | | General adult population (%) |
|---|---|---|---|
| | **N = 1000** | **%** | |
| **Sex** | | | |
| Women | 550 | 55.0% | 53.1% |
| Men | 450 | 45.0% | 46.9% |
| **Age groups (years)** | | | |
| 18–34 | 316 | 31.6% | 25.2% |
| 35–64 | 463 | 46.3% | 52.3% |
| 65 and over | 221 | 22.1% | 22.5% |
| **Highest education completed** | | | |
| Primary or less | 341 | 34.1% | 47.3% |
| Secondary | 363 | 36.3% | 32.2% |
| Tertiary | 296 | 29.6% | 20.5% |
| **Type of settlement** | | | |
| Budapest | 213 | 21.3% | 17.9% |
| Other cities | 557 | 55.7% | 52.6% |
| Village | 230 | 23.0% | 29.5% |
| **Region** | | | |
| Central Hungary | 348 | 34.8% | 30.4% |
| Eastern Hungary | 353 | 35.3% | 39.6% |
| Western Hungary | 299 | 29.9% | 30.1% |

Hungarian general population percentages refer to the population aged 15 years old and over; information on those data were based on the 2016 micro-census and made publicly available by the Hungarian Central Statistical Office. Primary level of education included those who had fully completed primary education or partly completed secondary education without direct access to post-secondary or tertiary education. Secondary level of education included those who fully completed secondary education or attended tertiary education without completing it. Tertiary level of education included those who had fully completed university studies.

education or less. Most respondents lived in cities (77%) and were evenly distributed across Hungary's regions (34.8% in Central, 35.3% in Eastern, and 29.9% in Western Hungary). Overall, the sample represented well the Hungarian general population in term of sex, age, type of settlement and region of residence. Respondents between 35–64 years old and people with lower educational levels were somewhat underrepresented. Further information about the respondents' characteristics can be found in a recent study that used the same sample [9].

Out of the 1000 respondents, 59% scored equal to or below 4 in a 7-point Likert scale on the degree of difficulty in answering the choice tasks. Conversely, 18% of the respondents scored the degree of difficulty equal to 7, i.e. the choice tasks were considered to be very difficult. When asked about which aspects contributed most for the degree of difficulty of the choice tasks, most respondents suggested that it was difficult to choose a preferred scenario (n = 555). Other reasons pinpointed by respondents were as follows: difficult to understand the differences between scenarios (n = 123), difficult to imagine that they needed medical care (n = 99), difficult to interpret the scenario's vignette (n = 48), or other reasons (n = 160).

## Preference weights for attributes of the care experience

As expected, the models in preference space suggest that the perceived utility of an outpatient consultation was greater when respondents faced shorter waiting times and positive experiences of care, all else equal (Table 3). The coefficients were all of the expected direction; also, estimates of both model 2 and 3 were fairly consistent in terms of magnitude. The constant term was negative and statistically significant across models. This may suggest that respondents were considering attributes not captured in the models or that there was "left-right bias", where respondents were more likely to choose consultation *A*.

Overall, respondents weighted attributes of the care experience with a doctor more in comparison with waiting time attributes. The attribute of the care experience that had the largest effect on respondents' preference across models was that of a *doctor spending enough time with a patient during consultation*. In model 1, where respondents were assumed to have the same preferences, the probability of a respondent choosing a consultation where a doctor spends enough time with a patient was 28% greater than that of choosing a scenario where a doctor did not, all else equal. This was followed by the attributes: *doctor giving opportunity to ask questions/raise concerns* and *doctor involving the patient in decision making about care/treatment* (marginal effect of 25%) and; a *doctor providing easy to understand explanations* (marginal effect of 14%). In addition, to wait an hour at a doctor's office had a larger negative effect on respondents' choice preference than that of an extra week of waiting time for an appointment.

The results of both mixed logit models in preference space suggested preference heterogeneity across attributes of the care experience, as indicated by statistically significant standard deviation coefficients; exception to this occurred for a *doctor spending enough time in the consultation* and a *doctor providing easy to understand explanations*. As an example, the respondents' preference for a consultation where a doctor spends enough time with a patient suggested that this attribute may yield twice as much utility (1.055 ÷ 0.438) as that when a doctor provides easy to understand explanations. In addition, model 3 showed significant heterogeneity in the preference for the out-of-pocket payment attribute. This provides evidence that model 2, where the out-of-pocket parameter was assumed to be fixed, was to some extent restrictive.

## Willingness to pay estimates

In model 4, we computed willingness to pay (WTP) estimates in the WTP space (Table 4). This model was analogous to model 3 in the preference space in that all attributes were

**Table 3. Estimates in preference space for conditional and mixed logit models.**

| Attribute | Conditional logit | | Mixed logit | | | |
| | Model 1 | | Model 2 | | Model 3 | |
| | Coef (SE) | Average (semi-) elasticity (SE) | Mean (SE) | SD (SE) | Mean (SE) | SD (SE) |
|---|---|---|---|---|---|---|
| $A_1$: Waiting time for an appointment (week) | – 0.065 *** (0.005) | – 0.031 (0.002) | – 0.146 *** (0.024) | 0.462 *** (0.207) | – 0.136 *** (0.014) | 0.214 *** (0.062) |
| $A_2$: Waiting time at the doctor's office (hour) | – 0.084 *** (0.014) | – 0.051 (0.007) | – 0.119 *** (0.021) | 0.200 *** (0.069) | – 0.160 *** (0.024) | 0.150 * (0.086) |
| $A_3$: Doctor spending enough time in consultation | 0.666 *** (0.048) | 0.281 (0.020) | 0.925 *** (0.067) | 0.007 (0.143) | 1.055 *** (0.078) | 0.237 (0.278) |
| $A_4$: Doctor providing easy to understand explanations | 0.268 *** (0.032) | 0.136 (0.017) | 0.403 *** (0.050) | 0.387 ** (0.146) | 0.438 *** (0.053) | 0.002 (0.657) |
| $A_5$: Doctor giving opportunity to ask questions/raise concerns | 0.382 *** (0.036) | 0.253 (0.017) | 0.570 *** (0.057) | 0.670 *** (0.098) | 0.622 *** (0.062) | 0.642 *** (0.111) |
| $A_6$: Doctor involving the patient in decision making about care/treatment | 0.425 *** (0.046) | 0.253 (0.020) | 0.567 *** (0.065) | 1.084 *** (0.090) | 0.726 *** (0.071) | 0.656 *** (0.124) |
| $A_7$: Out-of-pocket payment (€) | – 0.019 *** (0.001) | – 0.009 ($3.8 \times 10^{-4}$) | – 0.027 *** (0.001) | — | – 0.063 *** (0.011) | 0.200 *** (0.075) |
| Constant of choosing alternative $B$ | – 0.336 *** (0.039) | — | – 0.521 *** (0.064) | — | – 0.481 *** (0.068) | — |
| Log likelihood | – 2718.879 | | – 2627.843 | | – 2554.778 | |
| AIC | 5453.758 | | 5283.687 | | 5139.555 | |
| BIC | 5511.441 | | 5384.631 | | 5247.710 | |

* *p-value* < 0.05

** *p-value* < 0.01

*** *p-value* < 0.001; # Respondents = 1 000; # Observations = 10 000; Model 1: Conditional logit with dummy-variable coding; waiting times and out-of-pocket payment (€) were included as continuous variables. Model 2: Mixed logit with independent random and normally distributed coefficients for all attributes except out-of-pocket payment and alternative-specific constant (fixed effects); waiting time coefficients were given a log-normal distribution. Model 3: Mixed logit with independent random and normally distributed coefficient for all attributes and a fixed alternative-specific constant; waiting time and out-of-pocket payment coefficients were given a log-normal distribution. Attributes $A_3$ to $A_6$ were dummy-coded and the following were the base levels: 'The doctor does not spend enough time with you during the consultation' ($A_3$), 'The doctor explains things in a way that is not easy to understand' ($A_4$), 'The doctor does not give you an opportunity to ask questions or raise concerns about recommended treatment' ($A_5$) and 'The doctor does not involve you as much as you wanted to be in decisions about your care and treatment' ($A_6$). SE: Robust standard error; SD: Standard deviation; AIC: Akaike information criterion; BIC: Bayesian information criterion.

assumed to be independent, random and normally distributed, and waiting time and out-of-pocket attributes were given log-normal distributions. The means of the WTP measures varied from € 4.38 [95% confidence interval (CI): € 2.85 –€ 5.90] for the decrease of an hour in the waiting time at a doctor's office, to € 36.13 (95% CI: € 32.07 –€ 40.18) for a consultation where a doctor spends enough time with a patient relative to a consultation where a doctor does not spend enough time with a patient. Statistically significant standard deviations of WTP estimates suggest for relevant WTP heterogeneity across respondents, with exception to the attribute of a doctor spending enough time in consultation with a patient. Overall, respondents' WTP was larger for better care experiences than that for fewer waiting times. For improvement on the care experiences, respondents' WTP varied from € 15.61 (95% CI: € 12.38 –€ 18.84) for a doctor providing easy to understand explanations relative to when a doctor does not provide explanations in an understandable manner, to € 36.13 for a doctor spending enough time in consultation relative to a consultation where a doctor does not spend enough time with a patient. On the other hand, for improvement on waiting times, respondents' WTP varied from € 4.38 to wait an hour less at a doctor's office to € 5.46 (95% CI: € 4.02 –€ 6.90) for a week decrease on the waiting time for an appointment.

**Table 4. Mixed logit model in WTP space.**

| Attribute | Model 4 | | |
| --- | --- | --- | --- |
| | Mean (SE) | Median (SE) | SD (SE) |
| $A_1$: Waiting time for an appointment (week) | – 5.458 ** (0.736) | – 1.710 ** (0.280) | 16.545 *** (6.200) |
| $A_2$: Waiting time at the doctor's office (hour) | – 4.376 (0.779) | – 1.467 (0.479) | 12.303 *** (3.358) |
| $A_3$: Doctor spending enough time in consultation | 36.127 *** (2.069) | 36.127 *** (2.069) | 0.101 (3.645) |
| $A_4$: Doctor providing easy to understand explanations | 15.610 *** (1.649) | 15.610 *** (1.649) | 14.237 * (5.956) |
| $A_5$: Doctor giving opportunity to ask questions/raise concerns | 20.087 *** (2.022) | 20.087 *** (2.022) | 17.906 *** (5.006) |
| $A_6$: Doctor involving the patient in decision making about care/treatment | 21.876 *** (2.362) | 21.876 *** (2.362) | 38.627 *** (2.677) |
| $A_7$: Out-of-pocket payment (€) | – 0.065*** (0.018) | – 0.039 *** (0.005) | 0.085 *** (0.046) |
| Constant of choosing alternative B | – 18.357 *** (1.835) | — | — |
| Log likelihood | – 2611.5 | | |
| AIC | 5253 | | |
| BIC | 5361.156 | | |

* *p-value* < 0.05

** *p-value* < 0.01

*** *p-value* < 0.001; # Respondents = 1 000; # Observations = 10 000; Model 4: Mixed logit model in WTP space; model specifications are the same as in model 3 in preference space. Attributes $A_3$ to $A_6$ were dummy-coded and the following were the base levels: 'The doctor does not spend enough time with you during the consultation' ($A_3$), 'The doctor explains things in a way that is not easy to understand' ($A_4$), 'The doctor does not give you an opportunity to ask questions or raise concerns about recommended treatment' ($A_5$) and 'The doctor does not involve you as much as you wanted to be in decisions about your care and treatment' ($A_6$). SE: Robust standard error; SD: Standard deviation; AIC: Akaike information criterion; BIC: Bayesian information criterion.

The WTP estimates in the preference space using conditional logit (model 1) or mixed logit with the out-of-pocket payment coefficient fixed (model 2) were similar to those estimated in the WTP space (Table 5). Conversely, when the preference for the monetary attribute was allowed to be heterogeneous, the means of the WTP distribution estimated in preference space (model 3) seemed noticeably low across attributes in contrast with those estimated in the WTP space. Notwithstanding, the qualitative interpretation that respondents valued attributes of the care experience more than waiting time attributes holds.

## Discussion

### Main findings

This study undertook, to our knowledge, a novel approach with the use of standardized patient-reported experience measures (PREMs) to support a discrete choice experiment (DCE). We investigated the preference weights for attributes of the care experience in outpatient settings on a representative sample of the general adult population of Hungary. Also, the willingness to pay (WTP) for fewer waiting times and positive care experiences were analyzed, both in the preference and the WTP space.

Respondents preferred scenarios with better experiences of care and fewer waiting times. The care experience attribute with the largest effect on respondents' choice was that of a doctor

**Table 5. Comparison of willingness to pay estimates in preference and WTP space.**

| Attribute | Preference space | | | WTP space |
|---|---|---|---|---|
| | Model 1 | Model 2 | Model 3 | Model 4 |
| $A_1$: Waiting time for an appointment (week) | €3.50 (2.93–4.08) | €5.47 (3.82–7.12) | €2.16 (1.41–2.92) | €5.46 (4.02–6.90) |
| $A_2$: Waiting time at the doctor's office (hour) | €4.51 (2.98–6.03) | €4.46 (2.91–6.00) | €2.55 (1.59–3.50) | €4.38 (2.85–5.90) |
| $A_3$: Doctor spending enough time in consultation | €35.83 (30.98–40.67) | €34.68 (30.21–39.15) | €16.83 (11.26–22.40) | €36.13 (32.07–40.18) |
| $A_4$: Doctor providing easy to understand explanations | €14.43 (11.01–17.86) | €15.11 (11.62–18.59) | €6.98 (4.27–9.69) | €15.61 (12.38–18.84) |
| $A_5$: Doctor giving opportunity to ask questions/raise concerns | €20.55 (16.27–24.82) | €21.39 (17.10–25.68) | €9.92 (6.43–13.41) | €20.09 (16.12–24.05) |
| $A_6$: Doctor involving the patient in decision making about care/treatment | €22.86 (17.86–27.86) | €21.25 (16.58–25.91) | €11.57 (7.74–15.41) | €21.88 (17.25–26.51) |

In preference space, the willingness to pay was computed as the ratio of the estimated model coefficient for an attribute and the out-of-pocket payment coefficient. To compute estimates in WTP space we used Stata's user-written *mixlogitwtp* module. The 95% confidence interval (in brackets) were estimated with the delta method. Attributes $A_3$ to $A_6$ were dummy-coded and the following were the base levels: 'The doctor does not spend enough time with you during the consultation' ($A_3$), 'The doctor explains things in a way that is not easy to understand' ($A_4$), 'The doctor does not give you an opportunity to ask questions or raise concerns about recommended treatment' ($A_5$) and 'The doctor does not involve you as much as you wanted to be in decisions about your care and treatment' ($A_6$).

spending enough time with a patient. The second most preferred attributes on a consultation were those of being given the opportunity to ask questions/raise concerns about treatment and being involved in decision making. When the preference for out-of-pocket payments was allowed to be heterogeneous, the means of the WTP distributions estimated in the preference space and in the WTP space differed pronouncedly.

## Implications to the Hungarian health system

Our findings signal room for improvement on the responsiveness of the Hungarian health system to its citizen's expectations. First, a doctor spending enough time in consultation with a patient was found to be the most important aspect of the care experience; respondents, on average, were willing to pay the most for a positive experience on this attribute of the care experience relative to a consultation where a doctor does not spend enough time with the patient. These findings may be signaling an aspect of the health care system that is not sufficiently responsive to citizen's expectations and needs such as that of the time a doctor spends in consultation with a patient. In Hungary, the average length for a primary care consultation is of 6 minutes, rather short in contrast to that of other countries [26]. The extent to which this may affect the care experience and health outcomes is unclear. However, one's perception of consultation length is most often underestimated and confounded by their experiences of care (e.g. a patient that reports positive experiences of care with a consultation is likely to perceive longer consultation length) [27, 28]. Hence, our finding may in part be masking the need for improvement on other attributes of the care experience than that of the time a doctor spends in consultation with a patient. Aligning the resources of health care organizations to the expectations, needs and preferences of citizens, including those of patients, allow the health care system of becoming more patient-centered, with potential gains on health outcomes and patients experiences and satisfaction.

Second, involvement in decision making was an attribute of the care experience greatly valued by respondents. This is aligned with findings of a recent systematic review of DCE studies [13]. Shared decision making was also highlighted in other studies in Hungary, wherein data of our survey were reported: one presented that lesser positive experiences of care occurred regarding a patient being involved in decision making [9]; the other validated for Hungary a questionnaire on shared decision making [18]. In addition, recent evidence has suggested that, in Hungary, patients' preferences are less likely to be taken into account by GPs, in comparison

with other countries [29]. Hence, our findings seem to pinpoint that improvement is needed on empowering people who wish to undertake an active role in their health care management and be involved in decision making by their doctors.

Third, waiting time attributes represented a significant utility loss across models, but to a lesser extent compared with the remainder attributes of the care experience. Although waiting times account for a third of the unmet care needs in Hungary, and its effects on health outcomes are not documented in full [30], our findings hint that respondents are likely to overlook waiting time attributes. This could be the case for this specific group of respondents that, on average, seem to be willing to wait longer to receive care that might add value to aspects of the care experience that they prefer most, such as those related with the patient-doctor relationship and communication. However, similar evidence was found elsewhere [31], where attributes such as reputation and professional skills of a doctor or quality of the facilities weighted more on respondents' preferences than waiting time. This evidence might be supporting that waiting times in outpatient settings is not a pressing topic, at the moment, in Hungary.

## Preference heterogeneity

The model that fitted the data of our study better was that in preference space, where the preference for waiting times and out-of-pocket payments were allowed to be heterogeneous. The corresponding model in the WTP space seemed not to fit the data so well; however, the estimated WTP results were similar to those of more restrictive models in the preference space. Similar sensitivity to model specification was found in another study that used mixed logit models in preference and WTP space [23].

We observed preference heterogeneity across most attributes of the care experience, except that of a doctor spending enough time with a patient in consultation. This suggests that consultation length is of central importance to most respondents. Findings of a previous study, wherein the same sample was considered, suggest that experiences of care in regard to a doctor spending enough time in consultation with a patient varied significantly across respondents' characteristics (e.g. sex and age) [9]. Knowledge on the extent to which respondents' characteristics explain preference heterogeneity could be used to inform the decisions of policy makers in strengthening the responsiveness of the health care system via the implementation of quality assurance and improvement programmes that account for the citizens' voice. Given that citizens' expectations of care delivery evolve over time, in part because of previous experiences [32], it is paramount to have a comprehensive and fully functioning health system performance monitoring system in place, where capturing the perspective of patients and the general population is key.

## Strengths and limitations

Our study on eliciting preferences for care experiences in outpatient settings is strengthened for its large and representative sample, with no missing data in the DCE tasks. Also, the attributes of the DCE derived from a well-known international standardized set of PREMs which are widely adopted in health policy surveys, used for cross-national comparisons, and are relevant to citizens and policy-makers. Our findings should, however, be interpreted in light of study limitations. The method of survey delivery may have affected respondents' characteristics. This survey was online-based, which may have reduced the chance of participation to non-internet users and to people with low skills on information and communications technologies. These are usually characteristics of older people, which were reasonable represented in the study sample. Our findings are limited by the set of attribute levels considered. Whereas other attributes could have been considered, we chose those because of their relevance to

patient-centered care and the potential to allow future cross-national comparisons with other discrete choice experiments (e.g. benchmarking among OECD countries that collect patient experiences data with these PREMs). In our instrument design, we only considered main effects (to keep the number of choice sets at minimum in a relatively long survey) and have not investigated interaction terms in the regression models, given our choice for parsimonious models. Also, we did not include an opting-out option, which might have introduced some bias to estimates.

## Conclusions

This study contributes to the enhancement of our knowledge on the use of patient-reported experience measures (PREMs) to elicit people's preferences on attributes that shape the care experience. In Hungary, patient experience data collected via PREMs has thus far offered a static viewpoint into the performance of the health system in delivering value-based care. In this study, we provided evidence for the preference of a national representative sample for consultations where a doctor spends enough time with a patient and greater involvement in decision making. Moreover, respondents' willingness to pay for better experiences was greater than that for shorter waiting times. These findings could inform policy-makers and key-stakeholders on the value that citizens assign to aspects of the care experience, enhance actionability, and strengthen the monitoring of the health care system's responsiveness to citizens' needs and expectations.

In light of our methodological approach and findings, other studies could follow to explore the generalizability to other settings of care. Moreover, the understanding of the transferability of our findings to other countries may allow for cross-national comparison on what citizens value most regarding aspects of the care experience. A preference-based PREMs approach can inform the decisions of policy-makers, insurers, providers and other key-stakeholder to coordinate efforts and resource allocation in a more targeted manner. This could be achieved by prioritizing and acting on specific elements of the care experience that have a greater impact to the implementation of patient-centered care in a specific context and setting.

## Supporting information

**S1 Dataset. DCE module dataset.**
(XLSX)

**S1 File. DCE survey in Hungarian and translation to English.**
(PDF)

**S2 File. List of DCE choice sets organized by blocks.**
(XLSX)

## Acknowledgments

The authors would like to thank Kendall Gilmore, fellow of the Marie Skłodowska-Curie Innovative Training Network HealthPros (https://healthpros-h2020.eu), for his comments on an earlier draft of this manuscript. The authors extend their thanks to Armin Lucevic for reading and commenting the first draft of this manuscript.

## Author Contributions

**Conceptualization:** Óscar Brito Fernandes, Márta Péntek, Dionne Kringos, Niek Klazinga, László Gulácsi, Petra Baji.

**Data curation:** Óscar Brito Fernandes, Márta Péntek, Petra Baji.

**Formal analysis:** Óscar Brito Fernandes, Petra Baji.

**Funding acquisition:** Márta Péntek.

**Investigation:** Márta Péntek.

**Methodology:** Óscar Brito Fernandes, Petra Baji.

**Project administration:** Márta Péntek, László Gulácsi, Petra Baji.

**Supervision:** Márta Péntek, Dionne Kringos, Niek Klazinga, László Gulácsi, Petra Baji.

**Validation:** Óscar Brito Fernandes, Petra Baji.

**Visualization:** Óscar Brito Fernandes.

**Writing – original draft:** Óscar Brito Fernandes.

**Writing – review & editing:** Óscar Brito Fernandes, Márta Péntek, Dionne Kringos, Niek Klazinga, László Gulácsi, Petra Baji.

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
