## [Decision Letter · Decision Letter 0]

5 Dec 2019

PONE-D-19-29494

Eliciting preferences for outpatient care experiences in Hungary: A discrete choice experiment with a national representative sample

PLOS ONE

Dear Mr. Brito Fernandes,

Thank you for submitting your manuscript to PLOS ONE. After careful consideration, we feel that it has merit but does not fully meet PLOS ONE’s publication criteria as it currently stands. Therefore, we invite you to submit a revised version of the manuscript that addresses the points raised during the review process.

We would appreciate receiving your revised manuscript by Jan 19 2020 11:59PM. To enhance the reproducibility of your results, we recommend that if applicable you deposit your laboratory protocols in protocols.io, where a protocol can be assigned its own identifier (DOI) such that it can be cited independently in the future. For instructions see: http://journals.plos.org/plosone/s/submission-guidelines#loc-laboratory-protocols

We look forward to receiving your revised manuscript.

Kind regards,

Matthew Quaife

Academic Editor

PLOS ONE

Journal Requirements:

3. Please provide your institutional email address.

Reviewers' comments:

Reviewer's Responses to Questions

**Comments to the Author**

1. Is the manuscript technically sound, and do the data support the conclusions?

Reviewer #1: Partly

Reviewer #2: Partly

2. Has the statistical analysis been performed appropriately and rigorously? 

Reviewer #1: No

Reviewer #2: Yes

3. Have the authors made all data underlying the findings in their manuscript fully available?

Reviewer #1: Yes

Reviewer #2: No

4. Is the manuscript presented in an intelligible fashion and written in standard English?

Reviewer #1: Yes

Reviewer #2: Yes

5. Review Comments to the Author

Reviewer #1: The study has applied a DCE to address a topic of increasing relevance, namely the relative importance of the domains of PREMs. I have some comments to make that I trust would improve the manuscript.

1. For International readers it would be helpful to have a brief background to primary health care in Hungary. For example the order of magnitude for co-payments or out of pocket costs would help the reader put the WTP values into context. The WTPs for aspects of care for non threatening health issue seem very high.

2. Some attribute levels seem very high and fall way beyond a realistic scenario. For example 3 month wait for appointments, 94 euros cost, 4 hour wait time. Using such extremes could lead to biased estimates of attribute preference and hence WTP. Is timely access to primary health care an issue in Hungary?

3. It is not clear why the authors have chosen both conditional and mixed logit models. This should be justified in the methods section. The more complex mixed logit does not provide any additional information to support the findings and does not seem to be justified.

4. The method used for sub group analyses should be explained. I assume models were run on separate data sets? If so as noted in the limitation this is not an ideal approach as individual respondents will be present in multiple sub groups. E.g. age and gender, education and income. Given the large data set, the authors could have considered inclusion of interaction variable for major sub groups on a trial and error basis. Or again given the large sample size a latent class model would have been a better approach to addressing preference heterogeneity than sub group analysis. I would recommend that the authors consider alternate approaches.

5. The probability calculations and plot are confusing. The base case that has a 4% probability is clear, however what attribute levels are used to produce the curve is not clear. It maybe that I am not following the method. Irrespective in my view the probability calculations do not add to the findings. Rather the WTP provide a simpler estimate of attribute importance.

6. Table 1 should include comparison to general population distributions for Hungary as this was a stated aim of recruitment.

In summary, I suggest that the authors consider alternate approaches to addressing preference heterogeneity and respondent characteristics, for example using interaction variables or better still a latent class model.

Reviewer #2: 1. The methodology section would benefit from reorganisation to ensure a proper flow that readers can follow. The authors can learn from guidelines on how to report choice experiments such as Bridges et al. 2011 https://doi.org/10.1016/j.jval.2010.11.013

2. Deriving attributes and levels is a very important step of a choice experiment. The authors need to comprehensively explain in the text how they arrived to the selected attributes and levels. Were literature reviews and qualitative studies conducted? How did they reduce the number of attributes and levels (were experts engaged, were patients involved?). Explain these aspects in the text

3. The study lacks an opt-out or status quo Can the authors provide a very good justification in the text for excluding either one of these? This is because the lack of an opt-out or status quo exposes your WTP estimates to criticism.

4. What is the justification of the choosing the D-efficient design over other designs that exist such as Bayesian efficient designs? Which software was used to generate the experimental design? Researchers should explain these issues in the text.

5. How did the authors derive their prior parameters for the D-efficient design? Did they use educated best guess or were the priors derived from the pilot study they conducted? The authors need to explain this in the text. Furthermore, which model did the authors optimise for in the experimental design? MNL, MMNL etc

6. Line 73: Though the sample was obtained from a panel of an internet survey company, the authors need to be a bit detailed on how the quota sampling approach had been implemented to sample the respondents. Explain this in the text.

7. Line 172 “we assumed error terms to be independently and identically distributed following a logistic regression". I suggest the authors should replace the term "logistic regression" with "type 1 extreme value distribution".

8. Line 181 assuming the parameter of the out-of-pocket payment attribute as fixed rather than random is misleading. It suggests that the standard deviation of unobserved utility of the out-of-pocket attribute is the same for all observations. The authors should rerun their mixed multinomial logit model with the out-of-pocket payment attribute assuming a random and lognormal distribution instead of fixed.

9. Line 197-line 205. The authors should make it clear here that they computed the WTP measures in preference space using the conditional logit model coefficients. However, the authors still compute WTP estimates in preference space using the Mixed Multinomial Logit Model coefficients. They assume that the out-of-pocket payment attribute parameter is fixed and compute WTP as a ratio of parameters (-attribute/out-of-pocket) which is known as preference space. However, This can result in WTP distributions that are not well behaved as the authors are not accounting for variability in the price attribute (our-of-pocket payments). Therefore, the authors have to rerun the WTP estimates for mixed multinomial logit model in WTP space with the price parameter assuming a lognormal distribution. see Train and Weeks 2005 https://doi.org/10.1007/1-4020-3684-1_1

10. Attach the DCE questionnaire as supplementary file

6. PLOS authors have the option to publish the peer review history of their article (what does this mean?). If published, this will include your full peer review and any attached files.

Reviewer #1: Yes: Martin Robert Howell

Reviewer #2: No

---

## [Author Response · Author response to Decision Letter 0]

10 Feb 2020

Thank you for giving us the opportunity to submit a revised version of the manuscript entitled: “Eliciting preferences for outpatient care in Hungary: A discrete choice experiment with a national representative sample.” We greatly appreciate the Academic Editor's comments and those of the reviewers. We appreciate the time and efforts of the reviewers for their close review and thoughtful feedback; we feel that the paper has improved considerably by addressing their comments.

Comments from the Editor

E1. Please ensure that your manuscript meets PLOS ONE's style requirements, including those for file naming.

We proceeded accordingly.

E2. Please include additional information regarding the survey or questionnaire used in the study and ensure that you have provided sufficient details that others could replicate the analyses. For instance, if you developed a questionnaire as part of this study and it is not under a copyright more restrictive than CC-BY, please include a copy, in both the original language and English, as Supporting Information.

We provide as supporting information the DCE survey in its original language (Hungarian) and a translation to English (S1_File).

E3. Please provide your institutional email address.

We proceeded accordingly.

E4. We note that you have indicated that data from this study are available upon request. PLOS only allows data to be available upon request if there are legal or ethical restrictions on sharing data publicly. For information on unacceptable data access restrictions, please see http://journals.plos.org/plosone/s/data-availability#loc-unacceptable-data-access-restrictions.

We provide as supporting information a minimal anonymized dataset (S1_Dataset).

Comments from Reviewer 1

R1C1. For International readers it would be helpful to have a brief background to primary health care in Hungary. For example the order of magnitude for co-payments or out of pocket costs would help the reader put the WTP values into context. The WTPs for aspects of care for non threatening health issue seem very high.

We introduced a paragraph in the Introduction section to address this comment. It reads as follows:

“The Hungarian health system is organized around a single health insurance fund, which provides health coverage for nearly all residents. However, the benefit package is less comprehensive than in most European Union (EU) countries, and thus, a large number of people rely on out-of-pocket payments to access care [6]. Public health expenditure accounts for two-thirds of the total health expenditure, which sets the levels of out-of-pocket payments to almost double of the EU average (27% vs 16%) [6]. Out-of-pocket payments have been increasing partly because of the rising co-payments with pharmaceuticals and outpatient care, growing utilization of care providers in the private sector and the prevalence of informal payments [7]. Given this context, citizens’ experiences of care may be undermined up to some extent.” (Line 82–91)

We believe that with the given additional information an international reader may have a better understanding of the Hungarian context, especially regarding the relevance of increasing out-of-pocket payments in the Hungarian health care system. In addition, the references used in this paragraph may be of use for a reader who wishes more in-depth information about the Hungarian health system.

R1C2. Some attribute levels seem very high and fall way beyond a realistic scenario. For example 3 month wait for appointments, 94 euros cost, 4 hour wait time. Using such extremes could lead to biased estimates of attribute preference and hence WTP. Is timely access to primary health care an issue in Hungary?

We understand the concern of the Reviewer in regard to the breadth of some attribute levels, as they may seem unreasonable with the lenses of other contexts. However, they fit quite well to the Hungarian context. The health expenditure in Hungary is significantly below the EU average and only two-thirds are publicly funded. As such, Hungary has one of the highest levels of out-of-pocket payment in the EU and it represents almost twice as that of the EU average. In addition, the health benefit package is limited, which contributes to increasing out-of-pocket costs. Moreover, shortages and uneven distribution of health care professionals across the country undermine access to health services, especially outpatient services, which has been contributing for a growth in the utilization of private care services.

Given that there are very weak protection mechanisms in place to deal with the problem of increasing out-of-pocket payments, this is a great concern especially to most vulnerable populations. These were also the findings of a recent study that used the same data as we did in our DCE manuscript, where 27% of the respondents reported forgone medical visit due to difficulties in travelling, 24% unfilled prescriptions due to cost, 21% forgone medical appointments due to cost and 17 % skipped medical examinations due to costs (DOI: 10.1007/s10198-019-01063-0). A reporting on unmet medical needs because of waiting times is being prepared; preliminary findings were reported elsewhere (DOI: 10.1016/j.jval.2019.09.2097) and suggest that waiting time for an appointment or at a doctor’s office is common and frequently a problem to citizens.

R1C3. It is not clear why the authors have chosen both conditional and mixed logit models. This should be justified in the methods section. The more complex mixed logit does not provide any additional information to support the findings and does not seem to be justified.

Following the suggestion of the Reviewer, we have provided further information in the text to justify the use of both conditional and mixed logit models. The first two models (conditional logit and mixed logit with price attribute to be fixed) were included as benchmark model specifications and with the purpose of providing insights in regard to sensitivity testing using varying model specifications. Changes in text read as follows:

“Both model 1 and 2 were included as benchmark model specifications, where the latter is still quite common in the DCE literature because it allows the computation of willingness to pay estimates in preference space in a straightforward manner [20]. To improve the realism of model assumptions, in model 3 we have also specified the out-of-pocket coefficient to be log-normally distributed allowing the preference for this attribute to vary across respondents.” (Line 349–354)

R1C4. The method used for sub group analyses should be explained. I assume models were run on separate data sets? If so as noted in the limitation this is not an ideal approach as individual respondents will be present in multiple sub groups. E.g. age and gender, education and income. Given the large data set, the authors could have considered inclusion of interaction variable for major sub groups on a trial and error basis. Or again given the large sample size a latent class model would have been a better approach to addressing preference heterogeneity than sub group analysis. I would recommend that the authors consider alternate approaches.

After thoughtful consideration, we have decided to not report on these analyses. The Reviewer was correct to assume that the sub-group analyses were run on separate datasets. We now share similar concerns with this parsimonious segmentation approach, which may be very misleading. As suggested, we ran different model specifications with the inclusion of interaction variables in preference and WTP space. However, we faced several challenges in regard to computational power and the convergence of the models. Notwithstanding, we took note of the suggestion of using a latent class model, which we will consider in future studies.

R1C5. The probability calculations and plot are confusing. The base case that has a 4% probability is clear, however what attribute levels are used to produce the curve is not clear. It maybe that I am not following the method. Irrespective in my view the probability calculations do not add to the findings. Rather the WTP provide a simpler estimate of attribute importance.

After thoughtful consideration, we have decided to remove this component of the manuscript and re-focus the manuscript to WTP measures. Thank you.

R1C6. Table 1 should include comparison to general population distributions for Hungary as this was a stated aim of recruitment.

We added a column to Table 1 where we show the distribution of the general adult population of Hungary across the socio-demographic variables considered, for which we considered data from the micro-census held in 2016. All data were retrieved from the Hungarian Central Statistical Office. We removed the lines with information on the distribution of the sample by income tertiles from the Table, as these data will not be of use for the following sections. However, we provide a reference in the text where readers can access further information about the sample.

Comments from Reviewer 2

R2C1. The methodology section would benefit from reorganisation to ensure a proper flow that readers can follow. The authors can learn from guidelines on how to report choice experiments such as Bridges et al. 2011 https://doi.org/10.1016/j.jval.2010.11.013

We have made several changes throughout the manuscript, especially in the Methods section, to enhance the flow and reading experience of the manuscript.

R2C2. Deriving attributes and levels is a very important step of a choice experiment. The authors need to comprehensively explain in the text how they arrived to the selected attributes and levels. Were literature reviews and qualitative studies conducted? How did they reduce the number of attributes and levels (were experts engaged, were patients involved?). Explain these aspects in the text

We added further in text information about attribute selection. It reads as follows:

“The attribute selection for aspects of the care experience that add value to patients was based on the OECD’s proposed set of questions to gauge PREMs in outpatient settings [5]. Following best practices of attribute identification and selection [14], we chose those PREMs because of several reasons: 1) a recent systematic review often identified those measures in DCE studies to elicit patients’ preferences for primary health care [15]; 2) previous research has identified strong linkages between those attributes and quality of care, clinical safety and effectiveness [16, 17]; 3) those attributes are strong predictors of one’s perception of quality of an outpatient consultation [18], which may be an important consideration when choosing a consultation and; 4) those attributes represent a balance between what is relevant for patients and the health policy context [15]. Attributes covered aspects such as those of people’s access to care (e.g. waiting time for an appointment and waiting time at a doctor’s office) and experiences with outpatient care. About the latter, the attributes focused on aspects of care such as those of a doctor spending enough time with a patient, providing easy to understand information, giving a patient opportunity to ask questions or raise concerns about recommended treatments, and involving a patient in decision making about care and treatment. Additionally, we used a price attribute (out-of-pocket payments) to compose each outpatient consultation scenario (Table 1).” (Line 175–192)

In addition, we highlight that the PREMs statements were validated to the Hungarian language (process reported in a previous study DOI: 10.1007/s10198-019-01064-z) and that in the paper-based pilot of the DCE study the attributes and attribute levels were discussed with the participants.

R2C3. The study lacks an opt-out or status quo Can the authors provide a very good justification in the text for excluding either one of these? This is because the lack of an opt-out or status quo exposes your WTP estimates to criticism.

We have provided more detailed information in text on the reasoning for not including an opt-out or status quo option. It reads as follows:

“The DCE module started with a brief explanation about what was expected from the respondents regarding the choice tasks (S1 File). Afterward, respondents were instructed the following: “Imagine that you have a health problem that concern you but does not require immediate care and to receive health care you will be visiting a specialist for a consultation or an examination.” Next, respondents were asked to choose between two different outpatient consultation scenarios (A or B). All the tasks that were presented to the respondents for preference elicitation included all attributes, i.e. each consultation scenario was presented as full profile. We did not incorporate an opt-out or a status quo option. The inclusion of an opt-out option was not adequate given that in the task instructions provided to respondents we assumed that they would seek care because of a concerning health problem. Although an opt-out option might have reduced bias in parameter estimates, it would jeopardize a better understanding on respondents’ preferences if a large number of respondents choose the opt-out option. In addition, a status quo option was not included because this study aimed to estimate trade-offs between characteristics of a medical consultation (e.g. a doctor spending enough time in consultation with a patient or providing easy to understand explanations) rather than the expected uptake of certain consultations.” (Line 209–225)

R2C4. What is the justification of the choosing the D-efficient design over other designs that exist such as Bayesian efficient designs? Which software was used to generate the experimental design? Researchers should explain these issues in the text.

We are aware of the use of alternative approaches such as that of Bayesian efficient designs. However, that approach is not yet widespread as that of D-efficiency, as suggested elsewhere (DOI: 10.1007/s40273-018-0734-2), where the proportion of DCE studies using Bayesian D-efficiency was of 8% (n=23) in contrast with 35% (n=105) of those that followed a D-efficiency design. Hence, we have decided for an approach that is more familiar to readers of this type of studies and less technically demanding for one of our target audiences: policy-makers.

Following the comment of the Reviewer, we included the following in text:

“For the study to become feasible we defined a D-efficient fractional design with priors set to zero, for main effects only, with adequate level balance and minimum overlap of attribute levels. We used Stata’s dcreate command to maximize the D-efficiency of the design based on the covariance matrix of conditional logit model.” (Line 227–231)

R2C5. How did the authors derive their prior parameters for the D-efficient design? Did they use educated best guess or were the priors derived from the pilot study they conducted? The authors need to explain this in the text. Furthermore, which model did the authors optimise for in the experimental design? MNL, MMNL etc

We have provided more detailed information on the approach used. It reads as follows:

“For the study to become feasible we defined a D-efficient fractional design with priors set to zero, for main effects only, with adequate level balance and minimum overlap of attribute levels. We used Stata’s dcreate command to maximize the D-efficiency of the design based on the covariance matrix of conditional logit model.” (Line 227–231).

R2C6. Line 73: Though the sample was obtained from a panel of an internet survey company, the authors need to be a bit detailed on how the quota sampling approach had been implemented to sample the respondents. Explain this in the text.

We provided further information about the sampling approach in the text, as suggested. It reads as follows:

“The recruitment process aimed at a target sample size of 1000 respondents. A disproportionate stratified random sampling was employed to reflect the characteristics of the general adult population of Hungary in terms of sex, age (by age group: 18–24, 25–34, 35–44, 45–54, 55–64 or 65 and over years), highest education level attained (primary, secondary or tertiary), type of settlement (Budapest, other cities or village) and region of residence (Central, Eastern or Western Hungary). Given that this was an online survey and that the use of the internet is lower among people aged 65 and older, the sampling aimed to reflect a fair representativeness of older age groups in comparison with the distribution of older age strata in the general adult Hungarian population. We used publicly available information of the Hungarian Central Statistical Office to characterize the distribution of the general adult population [11].” (Line 120–131)

R2C7. Line 172 “we assumed error terms to be independently and identically distributed following a logistic regression". I suggest the authors should replace the term "logistic regression" with "type 1 extreme value distribution".

We agree with the suggestion of the reviewer and changed the text as suggested. It now reads as follows:

“We assumed errors to be independent and identically distributed following a type-one extreme value distribution.” (Line 337–338)

R2C8. Line 181 assuming the parameter of the out-of-pocket payment attribute as fixed rather than random is misleading. It suggests that the standard deviation of unobserved utility of the out-of-pocket attribute is the same for all observations. The authors should rerun their mixed multinomial logit model with the out-of-pocket payment attribute assuming a random and lognormal distribution instead of fixed.

We understand that assuming preference homogeneity for the out-of-pocket payment may be misleading and often unrealistic. However, considering a price attribute to be fixed is still a common approach to dealing with the challenges of computing WTP out of the ratio of two randomly distributed terms (DOI: 10.1007/s00181-011-0500-1). For this reason, we have decided to preserve the reporting of such model (model 2). Notwithstanding, we agree with the Reviewer that to account for heterogeneity in preferences we should have ran the analyses considering the out-of-pocket coefficient random and log-normally distributed. We accounted for this suggestion with model 3, as reported in the manuscript.

R2C9. Line 197-line 205. The authors should make it clear here that they computed the WTP measures in preference space using the conditional logit model coefficients. However, the authors still compute WTP estimates in preference space using the Mixed Multinomial Logit Model coefficients. They assume that the out-of-pocket payment attribute parameter is fixed and compute WTP as a ratio of parameters (-attribute/out-of-pocket) which is known as preference space. However, This can result in WTP distributions that are not well behaved as the authors are not accounting for variability in the price attribute (our-of-pocket payments). Therefore, the authors have to rerun the WTP estimates for mixed multinomial logit model in WTP space with the price parameter assuming a lognormal distribution. see Train and Weeks 2005 https://doi.org/10.1007/1-4020-3684-1_1

Our approach changed significantly, and we made changes throughout the text accordingly. We included Table 3 with the estimates in preference space with different model specifications (1: conditional logit; 2: mixed logit with out-of-pocket payment fixed and waiting times following a log-normally distribution and; 3: mixed logit with out-of-pocket payment and waiting times following a log-normally distribution). We also estimated a fourth model in WTP space with the same specifications of model 3 in preference space (Table 4). In addition, we included Table 5 where we contrast the WTP measures both in preference and WTP space.

R2C10. Attach the DCE questionnaire as supplementary file

We provide as supporting information the DCE survey in its original language (Hungarian) and a translation to English (S1_File).

---

## [Decision Letter · Decision Letter 1]

17 Apr 2020

PONE-D-19-29494R1

Eliciting preferences for outpatient care experiences in Hungary: A discrete choice experiment with a national representative sample

PLOS ONE

Dear Mr. Brito Fernandes,

Thank you for submitting your manuscript to PLOS ONE. After careful consideration, we feel that it has merit but does not fully meet PLOS ONE’s publication criteria as it currently stands. Therefore, we invite you to submit a revised version of the manuscript that addresses the points raised during the review process.

We would appreciate receiving your revised manuscript by Jun 01 2020 11:59PM. To enhance the reproducibility of your results, we recommend that if applicable you deposit your laboratory protocols in protocols.io, where a protocol can be assigned its own identifier (DOI) such that it can be cited independently in the future. For instructions see: http://journals.plos.org/plosone/s/submission-guidelines#loc-laboratory-protocols

We look forward to receiving your revised manuscript.

Kind regards,

Matthew Quaife

Academic Editor

PLOS ONE

Reviewers' comments:

Reviewer's Responses to Questions

**Comments to the Author**

1. If the authors have adequately addressed your comments raised in a previous round of review and you feel that this manuscript is now acceptable for publication, you may indicate that here to bypass the “Comments to the Author” section, enter your conflict of interest statement in the “Confidential to Editor” section, and submit your "Accept" recommendation.

Reviewer #1: All comments have been addressed

Reviewer #2: (No Response)

2. Is the manuscript technically sound, and do the data support the conclusions?

Reviewer #1: Yes

Reviewer #2: Partly

3. Has the statistical analysis been performed appropriately and rigorously? 

Reviewer #1: Yes

Reviewer #2: No

4. Have the authors made all data underlying the findings in their manuscript fully available?

Reviewer #1: Yes

Reviewer #2: Yes

5. Is the manuscript presented in an intelligible fashion and written in standard English?

Reviewer #1: Yes

Reviewer #2: Yes

6. Review Comments to the Author

Reviewer #1: I have reviewed the responses to my comments and those of the second reviewer and am satisfied that the authors have made appropriate changes to the manuscript to address these. I look forward to the publication.

Reviewer #2: I thank the authors for addressing some of the issues. However, after they rerun the models and revised their manuscript, some issues came up. I am still not satisfied with the manuscript.

1. The methods sections still doesn't flow yet. When reporting DCEs, the first thing in the methods section should be attribute and levels selection or study setting followed by attributes and levels. Your paper starts with how data were collected before outlining what was being collected. There already exists a checklist for reporting Health related DCEs e.g. Bridges 2011

You can rearrange your methods section as follows

1. Attribute selection [or study setting can come before attributes and levels]

2. Attribute levels selection

3. DCE tasks and Experimental design

4. Preference elicitation

5. Instrument design [This is what you call survey in your paper. Write about the sections of your questionnaire]

6. Data collection [ This is where you include the data collection section including ethical approval]

7. Statistical analyses

2. You aimed for sample size of 1000. What was this number based on? Rule of thumb, parametric method for calculating the sample size? State that in the text on how you derived the minimum sample size required.

3. The reason for excluding an opt-out or status quo is not convincing as in real market scenario a Hungarian patient can choose to opt-out of care or seeking care elsewhere. Though you assumed that the respondents would seek care, in your choice scenario, you did state that the health problem did not require immediate care. Patients should then have been allowed to opt-out as they can choose to delay care as it did not need immediate treatment. The opt-out could have possibly been labelled 'delay care' The repercussions for leaving out the opt-out is that it exacerbates hypothetical bias which further biases your WTP estimates. Therefore, you need to state this in your limitations. Furthermore, studies can still be designed in a way that includes and excludes the opt-out by for having two choice questions. For example, for those who choose the opt-out, you can ask them an additional question that forces them to choose between Alternative A or B only.

4. Line 198 "it would jeopardise a better understanding on respondents’preferences if a large number of respondents choose the opt-out option". This sentence would be better rephrased as "including the opt-out would not provide much information. However, this exposes the DCE to hypothetical bias as in real market scenario patients can opt-out of care or delay care.

5. Line 386. "Left bias". Did you mean "left-right" bias. In the model specification line 278, you state that where B0 is an alternative-specific constant that indicates respondents’ preference for consultation A vs. consultation B. But in your results tables 3,4,5 you define alternative specific constant as constant of choosing alternative B. This is a bit confusing. Is your alternative specific constant for Alternative A or Alternative B? Confirm

6. You have not stated in the statistical analyses section how you calculated your relative importance of attributes. You have provided information on how to calculated choice probabilities (preference weights) and WTP estimates but not relative importance estimates.

7. Tables 3,4,5, For the dummy coded variables, could you also state what the base levels were in the tables?

8. Line 465 "The standard deviation of the WTP estimates were high, suggesting a large preference heterogeneity across respondents" did you mean large heterogeneity in WTP estimates? Rectifying the wording used

9. By using the term ‘independent random’ throughout the text, do you mean ‘random and normally distributed’?

10. line 507 "To compute estimates in WTP space we used Stata’s user-written mixlogitwtp module". You had already mentioned this in the statistical analyses section

11. Check your interpretation of WTP estimates, they have to be relative to the base. e.g Patients were willing to pay.

12. Your discussion section is very weak. Authors need to be a bit detailed in discussing their findings in light of their research objectives and compare their finding with other studies in similar settings. Also, the authors should have some strong policy recommendations. They have attempted these, but it is too weak. The discussion section needs to be strengthened

Make these adjustments and let’s see how the manuscript looks.

7. PLOS authors have the option to publish the peer review history of their article (what does this mean?). If published, this will include your full peer review and any attached files.

Reviewer #1: Yes: Martin Howell

Reviewer #2: No

---

## [Author Response · Author response to Decision Letter 1]

1 Jun 2020

Comments from Reviewer 1

R1C1. I have reviewed the responses to my comments and those of the second reviewer and am satisfied that the authors have made appropriate changes to the manuscript to address these. I look forward to the publication.

Many thanks for your comments and time spent in reviewing our manuscript. We highly appreciate and value your contribution.

Comments from Reviewer 2

R2C1. The methods sections still doesn't flow yet. When reporting DCEs, the first thing in the methods section should be attribute and levels selection or study setting followed by attributes and levels. Your paper starts with how data were collected before outlining what was being collected. There already exists a checklist for reporting Health related DCEs e.g. Bridges 2011

You can rearrange your methods section as follows

1. Attribute selection [or study setting can come before attributes and levels]

2. Attribute levels selection

3. DCE tasks and Experimental design

4. Preference elicitation

5. Instrument design [This is what you call survey in your paper. Write about the sections of your questionnaire]

6. Data collection [ This is where you include the data collection section including ethical approval]

7. Statistical analyses

We rearranged the Methods section exactly as suggested by the Reviewer. We hope that the reading flows well now.

R2C2. You aimed for sample size of 1000. What was this number based on? Rule of thumb, parametric method for calculating the sample size? State that in the text on how you derived the minimum sample size required.

We clarified that the sample size of 1000 was based on rule of thumb. With a large sample size we could assign 250 people to each DCE block, and thus have sufficient confidence in model estimates. 

R2C3. The reason for excluding an opt-out or status quo is not convincing as in real market scenario a Hungarian patient can choose to opt-out of care or seeking care elsewhere. Though you assumed that the respondents would seek care, in your choice scenario, you did state that the health problem did not require immediate care. Patients should then have been allowed to opt-out as they can choose to delay care as it did not need immediate treatment. The opt-out could have possibly been labelled 'delay care' The repercussions for leaving out the opt-out is that it exacerbates hypothetical bias which further biases your WTP estimates. Therefore, you need to state this in your limitations. Furthermore, studies can still be designed in a way that includes and excludes the opt-out by for having two choice questions. For example, for those who choose the opt-out, you can ask them an additional question that forces them to choose between Alternative A or B only.

In our choice scenario, we assumed that respondents would seek care to their concerning health problem at some point in time, regardless of hypothetically opting for delaying care before visiting a doctor. Although an opt-out option could have reduced bias in model parameter estimates, given that it mimics better a real market scenario, we believe that for the purpose of our study, which was understanding the preference weights of attributes of the care experience, this bias is acceptable in contrast with the implications of having included an opt-out option. For example, including an opt-out option could have increased the choice task complexity, and thus, increased the proportion of respondents that opt-out because of this reason, which is specially concerning to less educated respondents.

Also, findings of a study (DOI: 10.1371/journal.pone.0111805) with the objective of determining to what extent the inclusion of an opt-out option in a DCE may have an effect on choice behavior found small differences between the forced and unforced choice model. This increased our confidence in not including an opt-out in this study. Notwithstanding, we are aware that studies can still be designed in a way that include and excludes the opt-out by for having two choice questions, as suggested by the Reviewer. We will consider this suggestion in future choice elicitation studies.

R2C4. Line 198 "it would jeopardise a better understanding on respondents’preferences if a large number of respondents choose the opt-out option". This sentence would be better rephrased as "including the opt-out would not provide much information. However, this exposes the DCE to hypothetical bias as in real market scenario patients can opt-out of care or delay care.

We followed the recommendation of the Reviewer and the revised the sentence as follows:

“We did not incorporate an opt-out or a status quo option, given that in the task instructions provided to respondents we assumed that they would have to seek care because of a concerning health problem at some point in time. Although an opt-out option could have reduced bias in parameter estimates, given that in real market scenario patients can opt-out of care or delay care, a forced choice method may lead to more thoughtful responses [1].”

R2C5. Line 386. "Left bias". Did you mean "left-right" bias. In the model specification line 278, you state that where B0 is an alternative-specific constant that indicates respondents’ preference for consultation A vs. consultation B. But in your results tables 3,4,5 you define alternative specific constant as constant of choosing alternative B. This is a bit confusing. Is your alternative specific constant for Alternative A or Alternative B? Confirm

We clarified this aspect in the text, following the comment of the Reviewer. 

R2C6. You have not stated in the statistical analyses section how you calculated your relative importance of attributes. You have provided information on how to calculated choice probabilities (preference weights) and WTP estimates but not relative importance estimates.

Mistakenly, we used the terms ‘preference weights’ and ‘relative importance’ interchangeably, which is not correct. In our analysis, we have not computed relative importance estimates. Throughout the text we clarified this aspect, removing all references to ‘relative importance’. Thank you for raising this issue.

R2C7. Tables 3,4,5, For the dummy coded variables, could you also state what the base levels were in the tables?

Thank you for this comment. Given that the attributes that were dummy coded (A3 to A6) can be easily interpreted as having a positive experience in that attribute relative to not having a positive experience, we have decided to leave the Tables as they stand. However, we added base level information in the footnotes of the Tables.

R2C8. Line 465 "The standard deviation of the WTP estimates were high, suggesting a large preference heterogeneity across respondents" did you mean large heterogeneity in WTP estimates? Rectifying the wording used

We rectified the wording.

R2C9. By using the term ‘independent random’ throughout the text, do you mean ‘random and normally distributed’?

We did mean ‘random and normally distributed’. Where applicable, we clarified this in the text.

R2C10. line 507 "To compute estimates in WTP space we used Stata’s user-written mixlogitwtp module". You had already mentioned this in the statistical analyses section

This information repetition is part of the footnotes to Table 4, so that readers who have only the time to quickly scan the manuscript’s Tables, can have a better understanding of the statistical analysis used to derive those estimates. 

R2C11. Check your interpretation of WTP estimates, they have to be relative to the base. e.g Patients were willing to pay.

Thank you for raising this issue. We have made changes throughout the text to bring clarity to the interpretation of the WTP estimates.

R2C12. Your discussion section is very weak. Authors need to be a bit detailed in discussing their findings in light of their research objectives and compare their finding with other studies in similar settings. Also, the authors should have some strong policy recommendations. They have attempted these, but it is too weak. The discussion section needs to be strengthened

We are sorry to know that following the previous rounds of review, the Reviewer still has this opinion in regard to the discussion of our findings. We believe that the discussion of our findings are aligned with the objectives of the study. However, we understand the critique of the Reviewer in regard to comparing our findings with those of other studies. To our knowledge, this is the first choice elicitation study focusing on aspects of the care experience using a set of standardized patient-reported experience measures to develop the choice tasks. We were not able to find a large number of elicitation studies to compare our results with; and those that we found were just too different and focusing on different measures (e.g. patient satisfaction). Partly because of sufficient evidence to contrast our findings with, we were cautious with the policy implications of our study. We do, however, pinpoint several policy implication to the Hungarian health care system, such as advocating for a wider involvement of citizens’ voice in setting the health agenda, having citizens’ and patients’ preferences partly steering health care organizations resources allocation, and supporting for systematic data collection of patients’ preferences and its use to inform the decisions of policy-makers encompassed in a broader health system performance monitoring and assessment framework.

In addition, we do believe that after our study, other could follow. Given that our choice sets were based on the statements of standardized PREMs, future cross-national comparisons are possible and desirable.

---

## [Editor Report · Decision Letter 2]

10 Jun 2020

Eliciting preferences for outpatient care experiences in Hungary: A discrete choice experiment with a national representative sample

PONE-D-19-29494R2

Dear Dr. Brito Fernandes,

We’re pleased to inform you that your manuscript has been judged scientifically suitable for publication and will be formally accepted for publication once it meets all outstanding technical requirements.

Kind regards,

Matthew Quaife

Academic Editor

PLOS ONE
---

## [Editor Report · Acceptance letter]

17 Jul 2020

PONE-D-19-29494R2 

Eliciting preferences for outpatient care experiences in Hungary: A discrete choice experiment with a national representative sample 

Dear Dr. Brito Fernandes:

I'm pleased to inform you that your manuscript has been deemed suitable for publication in PLOS ONE. Congratulations! Your manuscript is now with our production department. 

Kind regards, 

on behalf of

Dr. Matthew Quaife 

Academic Editor

PLOS ONE